# Innovative Approaches for Improving the Quality and Resilience of Spring Barley Seeds: The Role of Nanotechnology and Phytopathological Analysis

**DOI:** 10.3390/plants12223892

**Published:** 2023-11-18

**Authors:** Marzhan Sadenova, Natalya Kulenova, Sergey Gert, Nail Beisekenov, Eugene Levin

**Affiliations:** 1Priority Department Centre «Veritas» D. Serikbayev, East Kazakhstan Technical University, Ust-Kamenogorsk 070000, Kazakhstan; msadenova@edu.ektu.kz (M.S.); nkulenova@edu.ektu.kz (N.K.); bnail@edu.ektu.kz (S.G.); f23e503a@mail.cc.niigata-u.ac.jp (N.B.); 2Graduate School of Science and Technology, Niigata University, Niigata 950-2181, Japan; 3School of Applied Computational Sciences, Meharry Medical College, Nashville, TN 37208, USA

**Keywords:** spring barley, seed quality, disease resistance, fullerenols, nanopreparations, fusarium, mould

## Abstract

This study emphasizes the importance of seed quality in the context of yield formation. Based on the research data, this paper emphasizes the role of proper diagnosis of seed-borne pathogens in ensuring high and stable grain yields. Particular attention is paid to the study of the effect of the treatment of mother plants with fullerenol-based nanopreparations on the qualitative characteristics of spring barley seeds. The results showed that such treatment contributes to the increase in varietal purity, weight of 1000 grains as well as to the increase of nutrient and moisture reserves in seeds. Phytopathological analysis confirmed the presence of various diseases such as Alternaria, helminthosporiosis, fusarium, mold and mildew on the seeds. However, some samples showed a high resistance to pathogens, presumably due to the use of carbon nanopreparations. These results open new perspectives for the development of strategies to improve barley yield and disease resistance through seed optimization.

## 1. Introduction

Barley stands as a prominent agricultural crop globally. Within the Republic of Kazakhstan, the expanse of cultivated land surpassed 23 million hectares in 2022, with barley cultivation encompassing approximately 2.2 million hectares of this total, as referenced in [1]. As depicted in Figure 1 and corroborated by [2], Kazakhstan secured the 11th position among the world’s leading barley producers for the 2022–2023 period.

Barley grain contains the following ingredients: water—14%; protein—7–20%; carbohydrates—60–75%; fat—1.5–2.0%; fiber—5.5%; ash elements—2.5–3.0% [3]. One of the important indicators is ash content, which affects the quality of the resulting grain products. By determining the amount of ash, it is possible to determine the yield of flour during processing. Other factors have a strong influence on the quality of grain. The ability of crops to tolerate abiotic stresses such as excessive or insufficient water supply, high winds, extreme temperatures, frost, salt and other osmotic stresses is a key factor. In addition, climate change and extreme events increase the vulnerability of plants to pests and pathogens. Biotic factors also have a great influence on yield formation as they comprise living organisms in the environment such as plants, animals and microorganisms, especially specific diseases and pests.

The most important condition for the successful development of agricultural production is the strict control of product quality. Efforts for cleaner production are constantly increasing due to the environmental hazards caused by the intensive use of chemical fertilizers, especially N, P and K [4]. The use of chemical fertilizers forces plants to grow efficiently and quickly to meet food requirements. The disadvantages of using more chemical or synthetic fertilizers are environmental pollution, persistent changes in soil ecology, physical and chemical composition, decreased agricultural productivity and a number of health hazards. Climatic factors are responsible for increasing the abiotic stress of agricultural crops, which leads to a decrease in agricultural productivity [5]. Recently, much attention has been paid to the study of the possible effects of nanomaterials on plant growth and development [6]. One of these promising materials are carbon nanomaterials—fullerenes and their derivatives. Water-soluble fullerenols are new carbon-based nanomaterials with unique properties that allow them to find a wide application in agriculture. The most common nutritional disorder affecting plants is iron deficiency [7]. The authors studied the role of fullerenol [C_60_(OH)_22–24_] in the elimination of iron deficiency in *Cucumis sativus* with foliar treatment. It should be noted that studies of this kind have not been conducted in Kazakhstan. The use of water-soluble derivatives of fullerenes has not yet been widely spread in crop production due to the lack of sufficient knowledge about the mechanisms of their influence on agro- and ecosystems and their living components, including plants. Although, the prospect of using water-soluble fullerenes (fullerenols C_60_(OH)_22–24_ or C_70_(OH)_12–14_, fullerene C_60_ with glycine or with arginine: C_60_-L-Gly or C_60_-L-Arg) with different functional groups to enrich plant nutrition is already known [8]. The authors from [9] investigated both functionalized and non-functionalized carbon nanomaterials affecting fruit and crop yields. They showed that fullerene, C_60_ and carbon nanotubes increased the water-holding capacity, biomass and yield of plant fruits by about 118%, which is an outstanding achievement of nanotechnology in recent years. Fullerene-treated bitter melon seeds also increase the content of phytopreparations, such as cucurbitacin-B (74%), lycopene (82%), charantin (20%) and insulin (91%). Since just 50 µg/mL carbon nanotubes increase tomato production by about 200%, they can be used to increase agricultural production in the future. In [10], the possibilities of optimizing the production process and increasing the resistance of spring barley plants (*Hordeum vulgare* L.) to oxidative stress using fullerene derivatives C_60_- adducts with L-threonine, L-proline, L-hydroxyproline and L-histidine were studied. The positive effect of the fullerene C_60_ adduct with threonine on increasing the productivity and stability of wheat under drought conditions, as well as resistance to the disease caused by powdery mildew is known [11]. The most effective way of nanopreparation application is considered to be foliar application [12]. The technique of foliar application encompasses the delivery of nutrients directly to the leaves by spraying a solution containing one or more nutrients essential for plant development to be distributed to the other parts of the plant. This method is considered to be a quick and efficient means of overcoming plant nutritional deficiencies, as it supplies the plant with nutrients more easily than soil application (root absorption). Despite all the positive results of different authors from the use of fullerenes and fullerenols in crop production, Kole et al. note that extreme caution should be exercised, given the new knowledge about the accumulation and toxicity of nanoparticles in body tissues [13].

The basis of a good harvest is quality seed. The main indicators of seed quality are purity, weight of 1000 grains, water content in seeds (moisture), germination energy, laboratory germination and the data on the contamination of seeds with pathogenic microorganisms. The timely and correctly conducted phytopathological examination is of great importance, which allows to find out the presence and type of parasitic pathogenic organisms. The data on seed infestation by pathogenic microorganisms is one of the most important quality indicators, which allows to find out to what extent seeds are infected with parasitic and pathogenic organisms. Infected seeds either do not sprout or do not form full-grown plants, which results in lower yields. The main method of phytopathological examination is biological, in which crop seeds are germinated under optimal conditions. The biological method is designed to determine the external and internal infection of seeds. When analyzed, fungal and bacterial infections in the seed germinate together with the seed. According to the signs of infection, the species and quantitative determination of the pathogen in the seed sample is carried out. Microscopy and pure culture methods are used to accurately identify the species of the pathogen. The information obtained during the analysis allows a reliable assessment of the quality of crop seeds. In addition to the species composition of the seed pathogens present in the sample, the analysis also provides information on dead and abnormally sprouted plants, which will allow the seed rate to be adjusted.

The aim of this research is to investigate nanopreparations with a set of beneficial functions based on fullerenes for the treatment of crops in order to provide improved efficiency and manageability of agricultural production. It is important to determine the effect of fullerenol-based nanopreparations on spring barley yield, as well as to provide a phytopathological assessment of spring barley seed material.

## 2. Materials and Methods

### 2.1. Initial Data

Samples for this study comprised spring barley seeds, harvested in 2022 and obtained from plants treated with carbon-containing nanopreparations (Table 1). The area of the field experiment plot for F-1, F2 and F3 was 100 square meters with 4-fold repetition of variants (total treatment area was 400 square meters for each fullerenol species). Due to the limited amount of fullerenol adducts with manganese, cobalt and zinc, the total area treated was 100 square meters of each fullerenol species: F-4, F5 and F6. Barley crops not treated with fullerenol solutions served as a control sample. Spring barley crops were treated by spraying using hand-held sprayers. During the period of full maturity of spring barley, 5 samples were taken from an area of 1 square meter each in every study plot. Sampling was carried out on 8 August 2022.

All investigated nanopreparations with a complex of useful functions on the basis of fullerenes for treatment of crops with the purpose of providing the increase of efficiency and controllability of agricultural production were developed with the participation of the authors of this article. A number of articles [14,15,16,17] are devoted directly to the synthesis and identification of fullerenols.

Spring barley seeds, harvested in 2022 and obtained from plants not treated with the above-mentioned preparations served as the reference sample. The abbreviated name of this sample is «R».

### 2.2. Determination of Seed Purity

Seed purity is the weight quantity of pure seeds of the tested breed, expressed as a percentage of the total weight of seeds in the batch together with waste and impurities. Seed purity is one of the most important indicators used in determining the quality of seed material. It is known that impurities significantly reduce the quality of seeds in storage. Therefore, when assessing the quality of seeds, it is necessary to be very attentive to their purity, analyzing the composition of impurities and the degree of contamination in detail. Seed purity is determined in order to establish the weight content of normally developed seeds of the breed in the sample under study, as well as waste and impurities, and therefore in the batch it represents. When analyzing barley seeds, in order to separate stubby and small seeds, the sample was sifted through a sieve with straight-angled holes, 2.0 × 20 mm in size. Stubby seeds were additionally selected manually from the seeds remaining on the sieve. They are easily crumpled when pressed. The bulk of the seeds in the passage through the sieve and in the exit from the sieve was manually analyzed, identifying two main groups according to the standard: the seeds of the main crop and the waste consisting of various impurities.

Pure seeds included the following types of seeds:Whole, normally developed seeds regardless of their color;Small full-grained seeds that were equal to or more than half the size (length and thickness) of an average normally developed seed;Seeds that emerged and whose spines ruptured the seed coat but did not break through the seed coat;Seeds that were healthy in appearance but had cracked seed coats and where the embryo (endosperm) could not be seen through the cracks.Seed waste included the following fractions:Germinated seeds;Small seeds that were less than half the length and thickness of an average normally developed seed;Empty and flattened seeds, where the opposite walls of the shells were in contact with each other over the entire surface, regardless of their size;Mechanically damaged seeds (crushed, cut, broken with the embryo (endosperm), exposed and naked without the skin);Obviously rotten seeds, which changed their external color, or seeds that easily disintegrated when they were pressed with a spatula;Seeds affected by diseases (sclerotinia fungus, etc.);Seeds damaged by insects and mites;Seeds damaged by rodents.

The following fractions were classified as impurities:Seeds of trees and shrubs of other species;Seeds of agricultural crops and weeds;Seed pests, their larvae and pupae;Waste (lumps of earth, pebbles, sand, leaves, needles, cone scales, seed coats, excrement of rodents and insects, etc.).

After disassembly of the sample, clean seeds, waste and impurities were weighed (tolerable error not more than 0.01 g) on electronic analytical scales AUM210-E (“Abbota Corporation”, Chicago, IL, USA). Seed purity (in percentage) was determined by the ratio of the weight of pure seeds to the weight of the sample taken for analysis. Seed purity and the content of each fraction of waste and impurities were calculated with an accuracy of 0.01%.

### 2.3. Weight of 1000 Seeds

The mass of 1000 seeds is an indicator of the size and maturity of 1000 units of dry grains, expressed in grams. Determination of the mass of 1000 seeds allows to provide an assessment of nutrient reserves in seeds, i.e., the higher the mass of 1000 seeds of the same crop, the higher the content of nutrients in them. The mass of 1000 seeds was determined by a single sample according to standard methods described in [18]. To determine the mass of 1000 seeds of spring barley, the sample was weighed to a hundredth of a gram on electronic analytical scales AUM210-E (“Abbota Corporation”, Chicago, IL, USA), and recalculated manually. The value obtained by weighing the seed weight of the main crop was divided by the number of seeds and multiplied by 1000. Since the mass value of 1000 seeds was greater than 10 g, the result was determined to 0.1 g.

### 2.4. Seed Water Content (Moisture Content)

Seed moisture is the hygroscopic water content of the seed or the percentage of water in the seed sample at the time of sampling. This indicator affects seed quality and shelf life. At high moisture content, seeds are affected by bacteria, fungi and warming, and their germination rate is reduced. The moisture content of seeds is influenced by their ripeness, weather conditions during harvesting and storage conditions. Water enters seeds in two ways: through the parent plant—this is primary moisture—and from outside, through seed structures after physiological separation of the seed from the parent plant—this is secondary moisture. From the available spring barley seeds, samples were taken by crossing the jet of the pestered grains at the beginning, middle and end of their pestering. The seeds were ground in an agate mortar. From different places of the crushed mass, the samples were taken and each one was configured in pre-weighed bouquets, placed in a drying cabinet and heated to 130 °C. The seeds were dried for 40 min at the set temperature. At the end of drying, the bins were taken out of the desiccator, covered with lids and further cooled in the desiccator for 10–15 min. After cooling, the bouquets were weighed to the nearest hundredth of a gram on electronic analytical scales AUM210-E (“Abbota Corporation”, Chicago, IL, USA), and seed moisture content was calculated in percent using the following formula:(1)$$W=\left(\frac{\left(m-m^{1}\right)}{\left(m-m^{2}\right)}\right)\times 100\%$$
where *W*—moisture content of seeds, %; *m*—mass of the bunker with seeds before drying, g; *m*^1^—mass of the bunker with seeds after drying, g; *m*^2^—mass of empty bunker, g.

### 2.5. Laboratory Germination and Germination Energy

Phytopathological examination of seeds was carried out with the “roll method”. For laboratory analysis, the seed material was taken as a representative sample. For this purpose, seeds were laid out and distributed in an even layer so that the “thickness” of seeds was the same in all places. Two diagonal lines were then drawn across the seeds and ¼ of the total mass of the spread. The selected seeds were then spread and ¼ of this mass was sampled. A total of 2 samples of 20 seeds were counted from the second batch of selected seeds.

After selection, the seeds were spread with tweezers on double strips of 20 × 20 cm filter paper moistened with distilled water to full moisture content. Seeds were placed along the paper strip at a distance of 60–70 mm from the top edge and 1 cm apart, with the embryo facing away from itself. Photographs of spring barley seeds at each stage of preparation for phytopathological evaluation are presented in Figure 2 (a: image of threshed spring barley seeds after the drying process; b: image of spring barley seed preparation process for phytopathological evaluation with the “roll method”; c: image of spring barley seed preparation process for phytopathological evaluation with the “roll method”; d: image of spring barley seed during phytopathological evaluation with the “roll method”).

The vessel was placed in a thermostat at +20 °C and water was periodically added. Data were viewed and recorded on day 4 and day 10. During the analysis, the roll was spread out on a clean table treated with alcohol and the corex (top sheet of paper) was removed, freeing the seedlings for viewing.

The germination energy was determined on day 4 and calculated using the following Equation (2):(2)$$E=\frac{A\times 100\%}{a}$$
where *E* is the seed germination energy; *A* is the number of ungerminated seeds on the third day; *a* is the number of seeds set.

Seed germination was determined on the tenth day, when performing analysis using Equation (3) as follows:(3)$$P=\frac{n\times 100\%}{N}$$
where *P*—seed germination (%); *n*—number of germinated seeds on the tenth day; *N*—total number of plants in the sample.

### 2.6. Seed Contamination by Pathogenic Microorganisms

The timely detection of latent seed infection plays an important role in seed production. In this connection, the preliminary diagnosis of infected seeds is of great importance in improving seed quality, and is of the utmost importance along with the determination of seed germination. Before sowing, the seed material must be examined with a phytopathological examination to determine the percentage of infected seeds and the composition of fungal and bacterial phytopathogens. Based on the results of the phytopathological examination, the seed can be pre-treated with proven, modern and effective fungicides. This treatment helps to protect young seedlings from seed and soil infection. Phytopathological examination includes germination of seeds using the “roll method” and after 12–14 days, microscopic examination and identification of the species’ composition comprising pathogenic microorganisms. This procedure makes it possible to decide on the necessity of pre-sowing seed treatment, selection of the preparation method and its dosage for effective control of seed infection. The results of the phytopathological examination show which batch of seeds is best to use for seed purposes. By examining the seedlings under an optical research microscope BX-51 (OLYMPUS, Tokyo, Japan), the presence and species of a pathogen were noted. When diseases were recorded, one indicator (index) was determined: spread. Disease spread is calculated using Equation (4) as follows:(4)$$I=\frac{n\times 100}{N}$$
where *I*—percentage of infected plants (%); *n*—number of diseased plants in the sample; *N*—total number of plants in the sample.

## 3. Results and Discussion

### 3.1. Seed Purity Results

The results of determining the purity of spring barley seeds, harvested in 2022 and obtained from plants treated with fullerenol-based nanopreparation, as well as in the control sample, are shown in Table 2.

According to Table 2, the purest seed material of spring barley, harvested in 2022, is the sample obtained from plants treated with fullerenol adducts with zinc, cobalt, manganese and copper as well as in the control sample. An especially high value of varietal purity of spring barley is observed in the sample obtained from plants treated with a fullerenol adduct with manganese. This sample is the only sample superior in varietal purity in comparison to the control sample.

In conclusion, the purest sample of spring barley seed material, harvested in 2022, is the sample obtained from the plants treated with a fullerenol adduct with manganese. This sample is the only sample superior in varietal purity to the control sample.

### 3.2. Results of Determining the Weight of 1000 Grains

One of the most important elements of yield structure is grain size expressed through the weight of 1000 seeds. Of course, the weight of 1000 seeds is the second element of early productivity after the initiation of growth, but nutrient reserves, germination and seed viability are related to this parameter. This indicator is a “varietal trait” but can vary depending on growing conditions. The value of the weight of 1000 grains is an indirect indicator of “sufficient balance of the genetic complex” responsible for grain coarseness. As a rule, large grains have higher starch content, while small grains have higher protein content. The value of the weight of 1000 grains that is simple and relatively accessible for selection in early generations.

According to the researchers, this criterion is largely determined by the plant’s genotype and depends little on environmental changes. This fact allows to carry out “purposeful breeding work to increase barley productivity” [18,19].

The results of determining the weight of 1000 seeds of spring barley, harvested in 2022 and obtained from plants treated with nanopreparations based on fullerenols, as well as in the control sample, are shown in Table 3.

Thus, according to the data indicated in Table 3, it is possible to summarize the results obtained as follows: the comparative analysis of the weight of 1000 barley seeds obtained from plants treated with different fullerenol-based nanopreparations as well as the control sample without treatment are shown in the diagram in Figure 3.

The weight of 1000 grains of spring barley, harvested in 2022 and obtained from plants treated with carbon-containing nanopreparations is significantly higher compared to the control sample. The highest value of this indicator is observed in seeds obtained from plants treated with a fullerenol adduct with copper.

### 3.3. Seed Moisture Content Results

Barley storage is a complex technological process, which includes a number of measures. It is very important to create conditions in which the grain will retain all its quality characteristics for a long period of time. Storage methods may differ from each other; however, whatever the case, it is important to maximize the moisture content of the grain, as future yields will depend on this indicator.

Moisture movement inside the barley during storage is most often the cause of grain spoilage. Sometimes, the grain can be put into storage in an almost perfect condition, but there are some areas where the grain retains more moisture. In this case, the crop can also suffer completely. This is explained by the fact that changes in temperature can cause air currents that will carry moisture from one place to another. As a result, the grain starts to spoil everywhere.

To avoid this, it is important to control not only the moisture level of barley when placing it into storage, but also the temperature. Its index should not exceed the norm of +10 °C. Grains are better preserved at temperatures between +9 and +10 °C and a humidity of 12–13%—hence, barley can be stored for years. If the thermometer drops to +5 °C and humidity increases to 20%, barley will not survive in normal conditions for more than 3 months [20].

The maximum permissible humidity of normal barley should be 14% for storage up to 6 months and 13% for longer. Grain used for brewing purposes has greater requirements. Its moisture content should be 8 percent and 7 percent, respectively. The critical moisture value for barley is 15%. At this value and above, microorganisms begin to develop actively, and the grain begins to “breathe”. This leads to heat being is released, and the weight of the grain and its quality fall [21].

A comparative analysis of moisture content in spring barley seeds obtained from plants treated with various nanopreparations based on fullerenols, as well as the control sample without treatment, are shown in the diagram in Figure 4.

The moisture content of spring barley seeds, harvested in 2022, allows storage of this cereal for a sufficiently long time without a loss in grain quality. The majority of samples from plants treated with nanopreparations based on fullerenols, compared to the control sample, showed a higher water content in the seed structure.

### 3.4. Results of Germination Energy Determination

Germination energy characterizes seeds in terms of how quickly they germinate within a certain period of time. Germination energy is closely related to germination. The higher the germination energy and the smaller the difference between it and the germination of seeds, the better the quality of seeds. Reduced laboratory germination of seeds cannot be compensated by a corresponding increase in seeding rate; it is impossible to restore or replace poor seed quality, and it is possible to achieve the required number of plants per unit area, but not the full capacity of their growth and development. Seed germination is mainly influenced by meteorological conditions, especially during the period of grain formation and filling. For barley of the first class, the germination capacity should be not less than 95%, and not less than 90% for barley of the second class.

The image of the process of determining the germination energy on the 4th day after the beginning of the experiment of spring barley seeds is shown in Figure 5.

Figure 5 (F-1) shows spring barley grains from plants treated with individual fullerenol-24, while Figure 5 (F-4) shows those treated with a fullerenol adduct with manganese, with both of them carried out on the 4th day of the phytopathological examination with roll method The results of the determination of germination energy of spring barley seeds, harvested in 2022 and obtained from plants treated with fullerenol-based nanopreparation, as well as in the control sample, are shown in Table 4.

According to the results obtained, the control sample and seeds from the plants treated with fullerenol-d have the highest germination energy values. Seeds from the plants treated with fullerenol adducts with cobalt and zinc show the lowest values of germination energy. Also, it is worth noting that on the 4th day from the beginning of the experiment, spring barley grains from the plants treated with individual fullerenol-24 have the most developed shoots and root system compared to the other samples. The opposite effect is observed in spring barley grains from plants treated with a fullerenol adduct with manganese. The sample F-4, shown in Figure 4, has the least developed shoots and root system compared to the other samples.

### 3.5. Laboratory Germination Results

Determination of laboratory germination is one of the most important types of evaluations of seed sowing qualities as poor germination results in sparse crops, which reduces the yield. The germination rate should be close to 100%. The image of the process of determining laboratory germination on the 10th day after the beginning of the experiment of spring barley seeds is shown in Figure 6.

Figure 6 (F-4) shows spring barley grains from plants treated with a fullerenol adduct with manganese on the 10th day of the phytopathological examination with the roll method. The results of the determination of laboratory germination of spring barley seeds, harvested in 2022 and obtained from plants treated with fullerenol-based nanopreparation, as well as in the control sample, are shown in Figure 7.

According to the results obtained, the highest values of laboratory germination are observed in spring barley seeds from plants treated with a fullerenol adduct with zinc and fullerenol-d. The minimum values of laboratory germination are observed in the sample after treatment with a fullerenol adduct with cobalt. Minimum values of laboratory germination were observed in the sample after treatment with a fullerenol adduct with cobalt. 

It should be noted that the sample of spring barley grains from plants treated with a fullerenol adduct with manganese (Figure 6c), as well as during the registration of germination energy readings, conducted on the fourth day of the phytopathological examination and in comparison with other samples have the least developed shoots and root system.

Comparative analysis of germination energy and laboratory germination of spring barley seeds, harvested in 2022 and obtained from plants treated with nano-drugs based on fullerenols, as well as in the control sample, is shown in Figure 8.

Almost all samples of spring barley seeds show a significant decrease in the percentage of laboratory germination compared to germination energy. This may be due to the defeat of seeds by pathogenic microorganisms and their subsequent death. 

The only sample of spring barley seed material, in which the value of germination energy and laboratory germination are at the same level, is a sample of seeds from plants treated with a fullerenol adduct with zinc. This may be related to the acquired resistance of seed material to pathogenic microorganisms.

### 3.6. Results of Determination of Seed Pathogenic Microorganisms Infestation

Many pathogens are transmitted through seeds. Seeds are good nutrient substrates that are rich in proteins and minerals, which are essential for the normal activity of pathogens and bacteria.

Contaminated seeds can lead to significant yield losses and reduced grain quality. Seeds infected with dusty mildew of wheat and barley have reduced germination. *Fusarium* infections, *helminthosporiosis* root rot, *Alternaria* and bacteriosis can lead to death or damage of the root system of seedlings, which causes the thinning of crops. *Saprotrophic mould fungi* of genera such as *Penicillium*, *Aspergillus*, *Mucor*, *Rhizopus*, *Cladosporium*, *Epicoccum* and *Mucor* do not allow the normal development of young shoots and lead to their death. Dusty and hard heads of cereal crops and fusarium root rot are the cause of the reduction in productive stems. 

Young seedlings do not have a sufficient immune defense; therefore, some pathogens, which are in the soil, easily penetrate into young tissues. First of all, failure to observe crop rotation and the use of infected seeds leads to an increase in seed infection. Moreover, their accumulation in the soil occurs as a result of the use of chemical means of protection, which in turn leads to impoverishment of the microbiological composition of the soil. Such soils have a low antiphytopathogenic potential, which favors the accumulation of pathogens.

The results of determining the infestation of spring barley seeds with pathogenic microorganisms, harvested in 2022 and obtained from plants treated with nanopreparations based on fullerenols, as well as in the control sample, are shown in Figure 9.

In conclusion, the index of externally healthy seeds exceeds the values of grains infected with pathogenic microorganisms only in the control spring barley sample. Healthy seedlings can be seen in Figure 10.

Figure 10a shows the control sample, parallel No. I; (a1) control sample, parallel No. II; (Figure 10b) sample obtained from plants treated with a fullerenol adduct with copper; (Figure 10(b1)) sample obtained from plants treated with an individual fullerenol-24; (Figure 10c) image of a spring barley sample from plants treated with a fullerenol adduct with cobalt comprising a pink plaque of fusariotoxins.

According to the results, most of the seeds of the tested samples showed *Alternaria* spp. or the tan spot of wheat. The infected plants were underdeveloped when analyzed, and dark traces of the fungus sporulation were left on the filter paper (Figure 10b), which comprised an *Alternaria* spp. infection of spring barley seeds. The result of *Alternaria* spp. sporulation—dark spots on the filter paper—is shown in Figure 10b. Figure 10a shows a magnified image of *Alternaria* spp. as follows: 10(a2) 100-fold; 10(a3) 200-fold. Figure 10b shows a magnified image of *Helminthosporiasis* spp. (Bipolaris sorokiniana) as follows: 10(b2) 200-fold; 10(b3) 500-fold.

Spores are formed on short conidiophores with easily disintegrating spore chains that have longitudinal and transverse septa. Infestation during germination can lead to seedling death. *Alternaria* spores and hyphae can be seen in Figure 10a.

*Helminthosporiosis* (Bipolaris sorokiniana) as well as *Alternaria* were observed on all spring barley samples. A black color change was observed at the germinal end of the seed. *Helminthosporiosis* spores are shown in Figure 10b.

*Fusariosis* (Fusarium sp.) was observed only in the spring barley sample from plants treated with a fullerenol adduct with cobalt. The diagnostic sign of seedling fusarium is dark necrosis on spines and stems of a plant. Development of fusarium on seedlings leads to plant death. In the phytopathological examination, pink plaque is present on the filter paper, which can be removed by fusariotoxins. Fusarium damage is shown in Figure 10c.

The phytopathological analysis of each spring barley sample also revealed molds and spots that may be caused by *saprotrophic mold fungi* of the genera *Penicillium*, *Aspergillus*, *Mucor*, *Rhizopus*, *Cladosporium*, *Epicoccum* and others (Figure 11).

It is worth noting that compared to the control sample of spring barley, the indicators of the fungal disease infestation of seeds from plants treated with carbon nanopreparations are significantly lower. This fact may indicate the effectiveness of the impact of carbon nanopreparations.

## 4. Conclusions

Among the factors that play a major role in the formation of yield (fertilizers and chemical means of plant protection, seeds, machinery and technologies, natural and climatic conditions), the quality of seed material plays an important role. To obtain high and stable yields of grain crops, timely and correct diagnosis of seed-borne pathogens is of great importance. The examination and quality determination of spring barley grains obtained from mother plants treated with fullerenol-based nanopreparations showed an increase in varietal purity of seed material, as well as in the weight of 1000 grains, and consequently, an increase in the reserves of nutrients and moisture in seeds. This was also confirmed with a phytopathological examination of the seeds of grain crops, determining the presence and type of the pathogenic microflora transmitted by seeds. According to the phytopathological examination on spring barley seeds, harvested in 2022 and obtained from plants treated with carbon-containing nanopreparation, diseases such as *Alternaria*, *helminthosporiosis*, *fusarium* and mold were detected. However, certain samples showed pronounced resistance of seed material to pathogenic microorganisms, most likely acquired during the treatment of mother plants with the above-mentioned preparations.

Thus, methods for the synthesis of six types of water-soluble fullerenes derivatives (fullerenols) were developed, and experimental batches of each product were obtained. The positive effect of carbon-containing nanomaterials (fullerenes and their water-soluble derivatives—fullerenols) in crop cultivation is confirmed by the authors in [7,22,23]. Their application helps to increase crop yields by stimulating water retention and controlling certain pathogens [24,25]. Studies presented in [26,27] show that fullerenol regulates oxidative stress and tissue ion homeostasis in spring wheat, improving its net primary productivity under salt stress. Studies on the structure, properties and other characteristics of fullerenols have been continued and presented in [28,29,30]. Our future publications will present the results of testing synthesized nanopreparations in field conditions on spring barley crops under the conditions of three different soil-climatic zones of eastern Kazakhstan.

## Figures and Tables

**Figure 1 plants-12-03892-f001:**
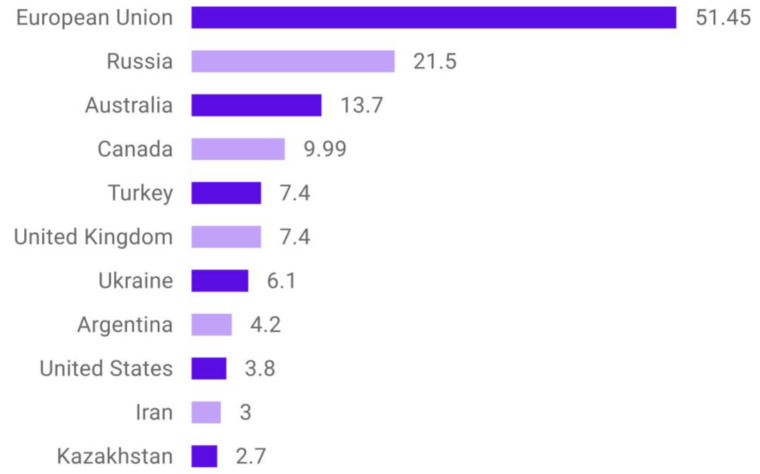
Major barley producers worldwide in 2022–2023, by country in million metric tons).

**Figure 2 plants-12-03892-f002:**
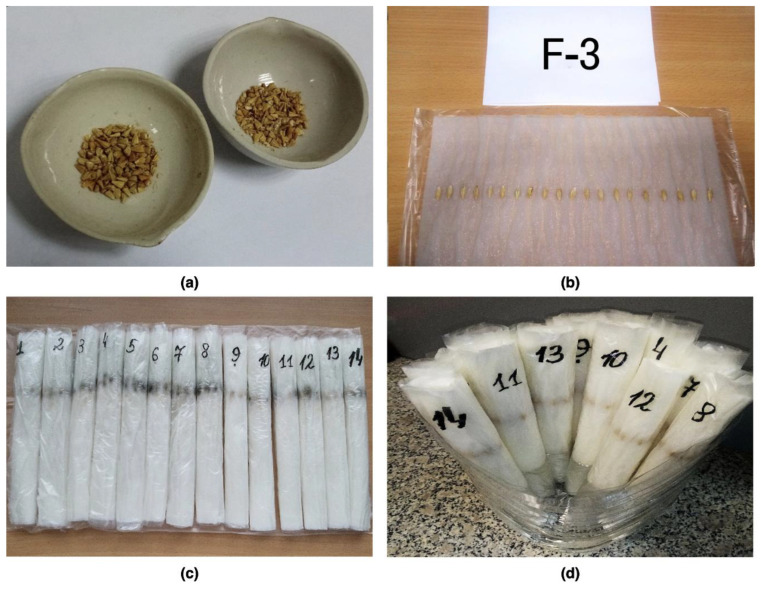
Photographs of spring barley seed material at each stage of preparation for phytopathological evaluation: (**a**) threshed seeds after moisture determination; (**b**) arrangement of seeds on filter paper; (**c**) sam-ples for examination; (**d**) seed germination stage.

**Figure 3 plants-12-03892-f003:**
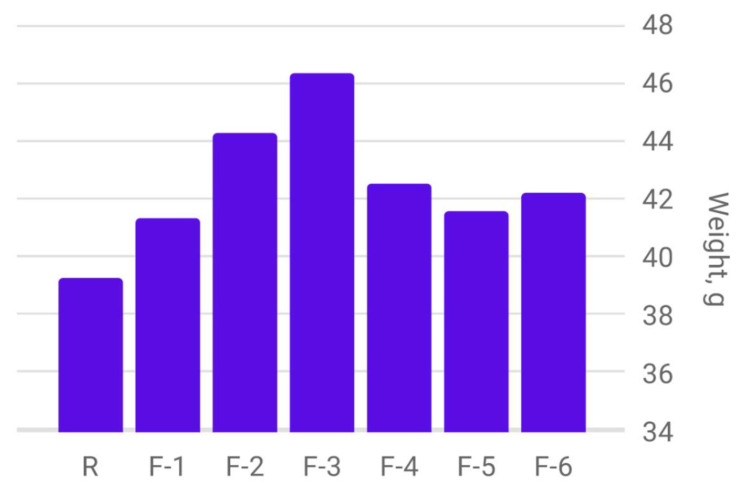
Diagram of the determination of the weight of 1000 grains of spring barley depending on the treatment with carbon nanopreparations.

**Figure 4 plants-12-03892-f004:**
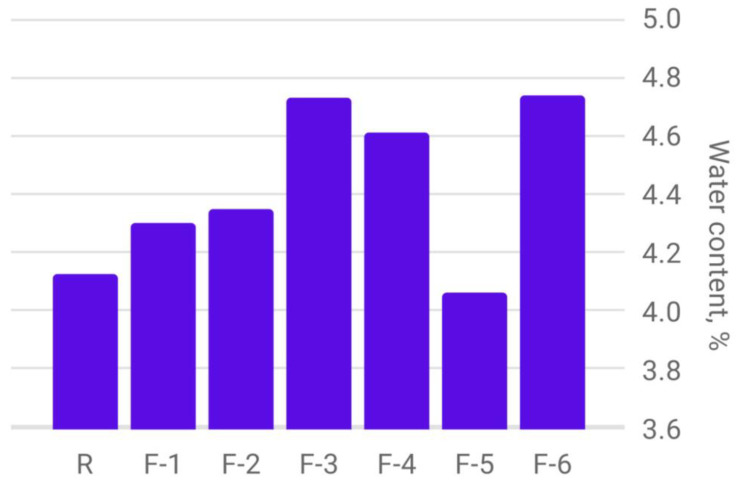
Diagram of the determination of the weight of 1000 grains of spring barley depending on the treatment with carbon-containing nanopreparations.

**Figure 5 plants-12-03892-f005:**
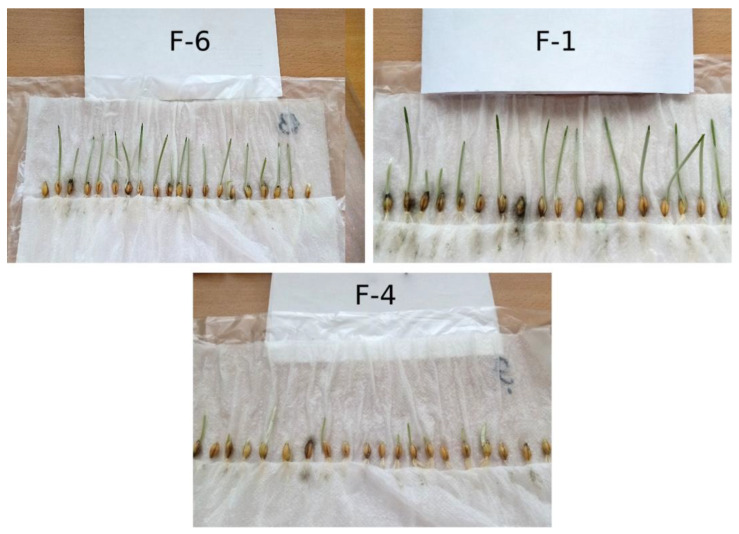
Process of determination of germination energy of treated spring barley seeds on the 4th day of phytopathological examination with the roll method.

**Figure 6 plants-12-03892-f006:**
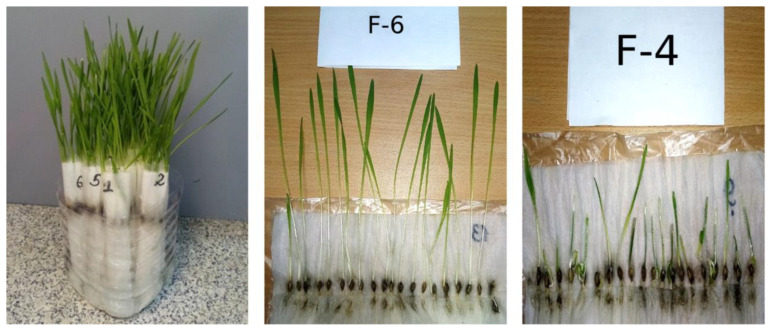
Process of determining laboratory germination of spring barley seeds.

**Figure 7 plants-12-03892-f007:**
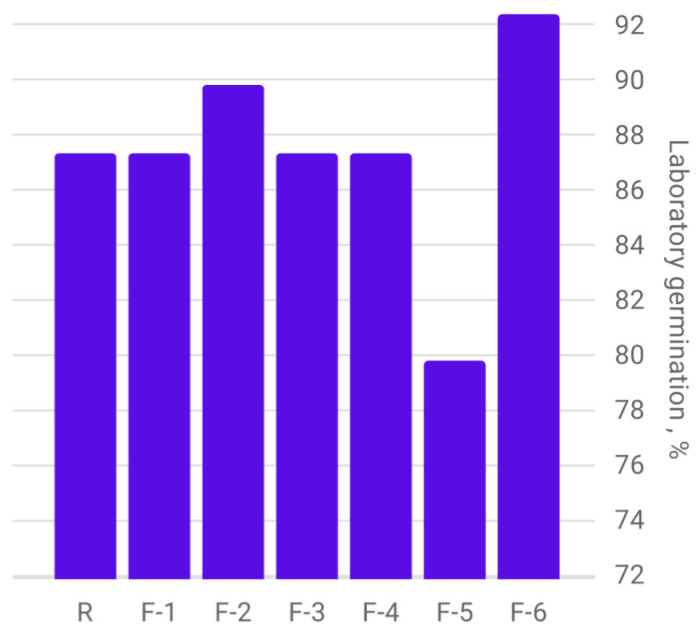
Diagram of the determination of laboratory germination of spring barley seeds depending on the treatment with carbon-containing nanopreparations.

**Figure 8 plants-12-03892-f008:**
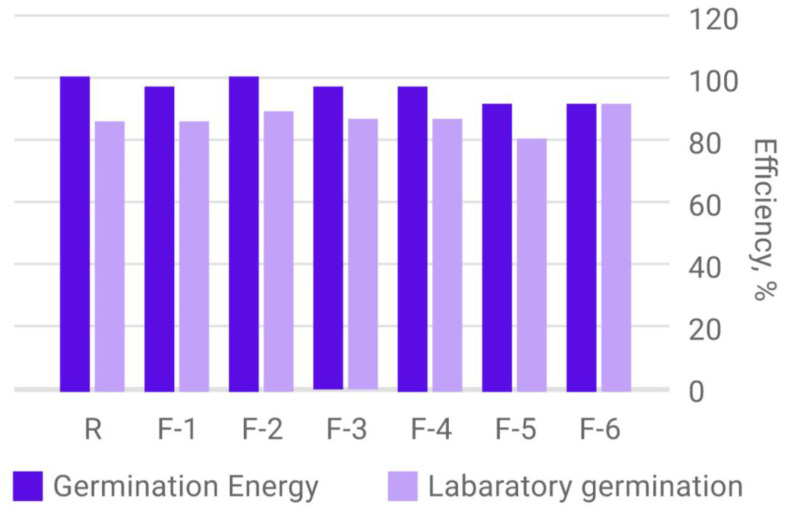
Diagram of the determination of germination energy and laboratory germination of spring barley seeds depending on the treatment with carbon-containing nanopreparations.

**Figure 9 plants-12-03892-f009:**
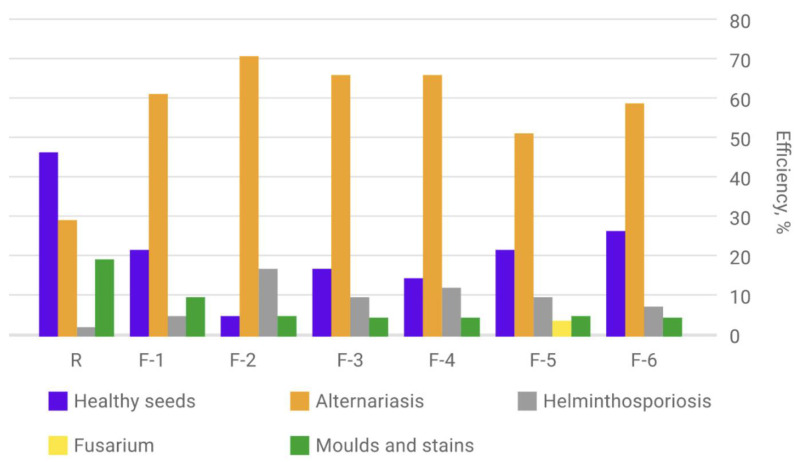
Diagram of the results of determining the contamination of spring barley seeds with pathogenic microorganisms, harvested 2022 and obtained from plants treated with nanopreparations based on fullerenols, as well as in the control sample.

**Figure 10 plants-12-03892-f010:**
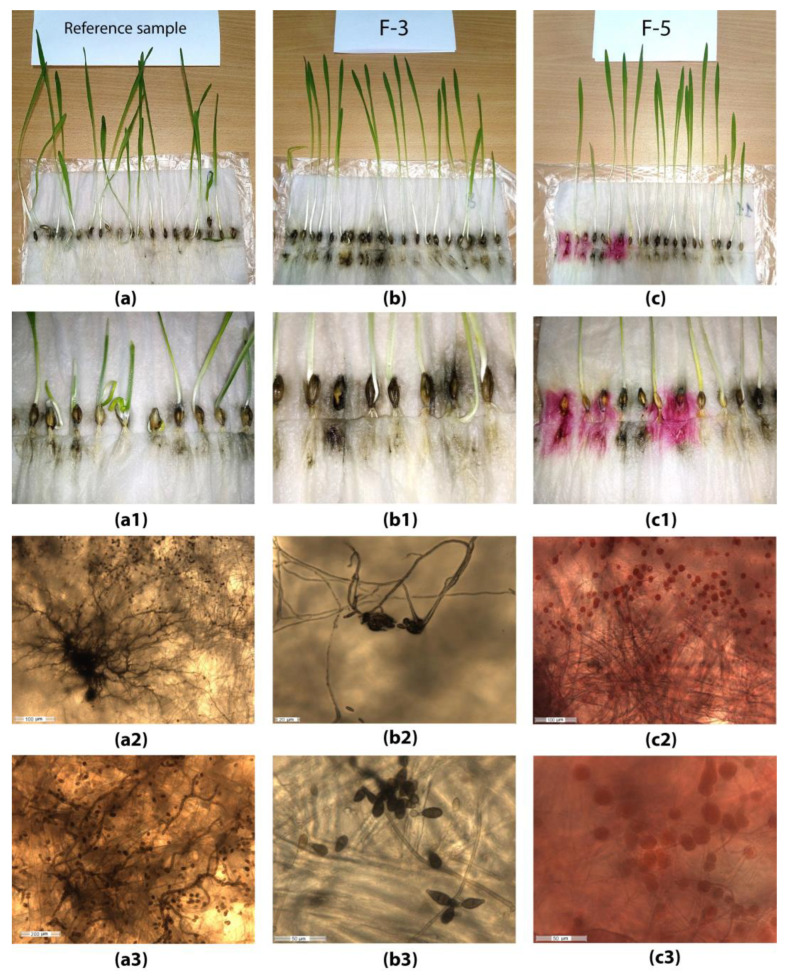
The process of determining pathogenic microorganism infestations of spring barley seeds: (**a**) *Alternaria* spp. in control sample (parallel № 1); (**a1**) *Alternaria* spp. in control sample (parallel № 2); (**a2**) *Alternaria* spp. at 100-fold magnification; (**a3**) *Alternaria* spp. at 200-fold magnification; (**b**) *Bipolaris sorokiniana* in a seed sample from plants treated with F-3; (**b1**) *Bipolaris sorokiniana* in a seed sample from plants treated with F-1; (**b2**) *Bipolaris sorokiniana* from seeds at 200-fold magnification; (**b3**) *Bipolaris sorokiniana* from seeds at 500-fold magnification; (**c**) *Fusarium* sp. In a seed sample from F-5-treated plants (parallel №1); (**c1**) *Fusarium* sp. in a seed sample from F-5-treated plants (parallel № 2); (**c2**) *Fusarium sp.* from seeds at 200-fold magnification; (**c3**) *Fusarium sp.* from seeds at 500-fold magnification.

**Figure 11 plants-12-03892-f011:**
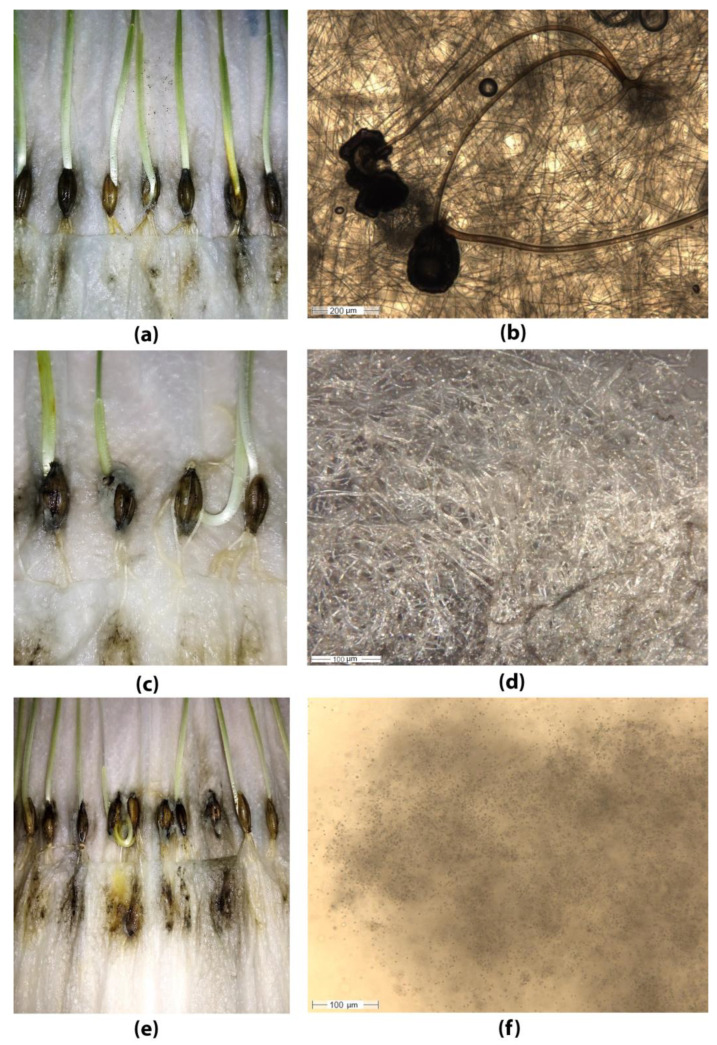
Saprotrophic mold fungi identified with phytopathological analysis of barley seeds: (**a**,**b**) genus Rhizopus; (**c**,**d**) genus Mucor; (**e**,**f**) genus Aspergillus.

**Table 1 plants-12-03892-t001:** Abbreviations for the corresponding nanopreparations.

No. n/a	Fullerenol Abbreviation	Full Name of Fullerenol
1	F-1	Personalised fullerenol-24
2	F-2	Fullerenol-d
3	F-3	Adduct of fullerenol with copper
4	F-4	Fullerenol adduct with manganese
5	F-5	Fullerenol adduct with cobalt
6	F-6	Fullerenol adduct with zinc

**Table 2 plants-12-03892-t002:** Calculation of varietal purity in a control sample of spring barley seeds (R).

Type of Treatment for the Calculation of Varietal Purity in a Spring Barley Seed Sample Obtained from Plants	Group Name	Attachment No.									
		1st Attachment		2nd Attachment		3rd Attachment		4th Attachment		Average Value, %	Grade Purity Categories
		g	%	g	%	g	%	g	%		
Control sample of spring barley seeds (R)	Seeds of the main crop	3.47	96.76	2.37	95.39	2.07	95.14	2.07	94.55	95.46	III
	Waste	0.11	3.23	0.11	4.60	0.10	4.85	0.11	5.44	4.53	
Individualized fullerenol-24 (F-1)	Seeds of the main crop	2.89	89.46	2.25	93.32	2.97	96.45	2.65	96.49	93.93	-
	Waste	0.34	10.53	0.16	6.67	0.10	3.54	0.09	3.50	6.06	
Fullerenol-d (F-2)	Seeds of the main crop	2.14	92.41	3.19	96.14	3.18	94.83	3.94	95.69	94.77	-
	Waste	0.17	7.58	0.12	3.85	0.17	5.16	0.17	4.30	5.22	
Fullerenol adduct with copper (F-3)	Seeds of the main crop	4.05	95.20	3.02	93.11	3.20	96.66	4.58	96.72	95.42	III
	Waste	0.20	4.79	0.22	6.88	0.11	3.33	0.15	3.27	4.57	
Fullerenol adduct with manganese (F-4)	Seeds of the main crop	3.21	97.51	-	-	-	-	-	-	97.51	III
	Waste	0.08	2.48	-	-	-	-	-	-	2.48	
Fullerenol adduct with cobalt (F-5)	Seeds of the main crop	3.06	95.31	-	-	-	-	-	-	95.31	III
	Waste	0.15	4.68	-	-	-	-	-	-	4.68	
Fullerenol adduct with zinc (F-6)	Seeds of the main crop	3.98	95.02	-	-	-	-	-	-	95.02	III
	Waste	0.21	4.97	-	-	-	-	-	-	4.97	

**Table 3 plants-12-03892-t003:** Determination of the weight of 1000 grains in a control sample of spring barley seeds (R).

Type of Treatment for the Calculation of Varietal Purity in a Spring Barley Seed Sample Obtained from Plants Treated	Number of Grains in the Sample, pcs.	Sample Weight, g	Average Weight of 1 Grain, g	Average Weight of 1000 Seeds, g
Control sample of spring barley seeds (R)	254	9.94	0.0391	39.1
Individual fullerenol-24 (F-1)	261	10.75	0.0412	41.2
Fullerenol is d (F-2)	281	12.4	0.0441	44.1
Adduct of fullerenol with copper (F-3)	320	14.83	0.0463	46.3
Adduct of fullerenol with manganese (F-4)	75	3.19	0.0425	42.5
Adduct of fullerenol with cobalt (F-5)	73	3.04	0.0416	41.6
Adduct of fullerenol with zinc (F-6)	94	3.97	0.0422	42.2

**Table 4 plants-12-03892-t004:** Results of the determination of germination energy of spring barley seeds, harvested in 2022 and obtained from plants treated with fullerenol-based nanopreparations, as well as in the control sample.

Sample Name	Germination Energy, %			Class
	Sample No. I	Sample No. II	Total	
R	100.0	100.0	100.0	I
F-1	100.0	95.0	97.5	I
F-2	100.0	100.0	100.0	I
F-3	95.0	100.0	97.5	I
F-4	95.0	100.0	97.5	I
F-5	90.0	95.0	92.5	II
F-6	95.0	90.0	92.5	II

## Data Availability

Data are contained within the article. The data presented in this study are available from the corresponding author upon reasonable request.

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
