# Peer review of "Innovative Approaches for Improving the Quality and Resilience of Spring Barley Seeds: The Role of Nanotechnology and Phytopathological Analysis"

_plants, 2023, doi:10.3390/plants12223892_

Round 1

Reviewer 1 Report

Comments and Suggestions for Authors

Author Response

Thank you very much for your valuable review.

Our response is in the attached file.

Reviewer 2 Report

Comments and Suggestions for Authors

Manuscript Innovative "Approaches to Improving Quality and Resilience of Spring Barley Seeds: The Role of Nanotechnology and Phytopathological Analysis" needs correction before the next stage of the publication process.

lines 31-33 Sentence must be improved

line 37 A range is given for most ingredients of barley grain. Why there is one value for ash content. Does it change under the influence of biotic and abiotic factors?

Please check and correct fullerenol chemical formula instead f.ex. C60(OH)22-24 should be C60(OH)22-24 (line 51 and other citation of fullerenol in the manuscript text).

line 52 (and others) species scientific (vascular plants and microorganisms) name must be written in Italic font.

Additional information is needed about treatment of barley plants with carbon nanoparticles. The authors reported only that barley plants were harvested in 2022. There is no information about the  fullerenol application term, application method, size of the sample harvested or date of the sample taken.

The methodology requires major changes. The authors provide a lot of unnecessary information or the description of the features should not affect the use of fullerenol. For example, the purity of grain depends mainly on the cleaning management methods. 

In other descriptions, I suggest omitting some information. For example:

2.3 Weight of 1000 seeds 

The mass of 1000 seeds is an indicator of the size and maturity of 1000 units of dry grains, expressed in grams. Determination of the mass of 1000 seeds allows to give an assessment of the reserves of nutrients in seeds, i.e. the higher the mass of 1000 seeds of the same crop, the higher the content of nutrients in it.

or

2.4 Seed water content (moisture content) 178

Seed moisture is the hygroscopic water content of the seed or the percentage of water in the seed sample at the time of sampling. This indicator affects seed quality and shelf  life. At high moisture content, seeds are affected by bacteria, fungi and warming, and their germination rate is reduced. The moisture content of seeds is influenced by their ripeness, weather conditions during harvesting and storage conditions. Water enters seeds in two ways: through the parent plant - this is primary moisture and from outside, through seed structures after physiological separation of the seed from the parent plant - this is secondary moisture.

This is not information regarding the description of the research methods used.

Symbols in formula 2 is not connected with formula description (line 224-225).

In formula 2 Authors counted ungerminated seeds and P (or E) not determined germination energy.

Formula 3. Comments the same: Authors counted ungerminated seeds and P not determined seed germination.

Formula 4. Instead disease distribution my suggestion is percentage of infected plants.

Explanation need information presented in table 2. Why for F-4, F-5 and F-6 only one data presented. Lack of explanation in the manuscript.

Line 257 Instead table 3 should be table 2.

Line 260-266. Seeds purity repeated and resented two time the same value. I suggest add one column in table 2 with seed purity categories.

The data presented in table 3 suggest that the research based on 1 plant (maybe even on 1 barley ear)? Additional average weight of 1000 grains need correction. F.ex. for F-1 weight of 1000 grains is 39.22 g. Basic on data of grain number and sample weight average 1000 seeds weight should be 38.86 g. 

Author Response

(The authors gave the same response as above.)

Reviewer 3 Report

Comments and Suggestions for Authors

This paper studied the influence of fullerenol-based nanopreparations to spring barely seeds and the role of correct diagnosis of seed-borne pathogens in ensuring high and stable grain yields. There are some problems should be solved before it is considered for publication, which are listed below.

1. The keywords are too many, please moderate cut.

2. Page 2, L35. In the note of Figure 1: “2022/2023” should be changed to “2022-2023”.

4. Page 2, L40-47. The pesticide described in this paragraph are not related to nutrients lacking in plants, this paragraph is not logical, please adjust.

5. Page 3, L100. This sentence should put the test subject.

6. Page 5, in the 2.3 Weight of 1000 seeds, there is no indication of whether there are duplicates and how many duplicates there are.

7. Page 6, Figure 5 is not aesthetically done, the picture should be taken from above.

8. Page 10, L304. There is no first line indent.

9. Page 10, L312. The meaning of “+9...+10°C” is not clear.

10. Page 10, L311. Change “+10 ° C” to “+10°C”. Please check the full text carefully.

11. Page 15, L404. There is no first line indent. Please check the text carefully.

12. Page 15, L417. Change “Epicoccum, Mucor” to “Epicoccum and Mucor”.

13. Page 15, L418. Change “cereal crops, fusarium” to “cereal crops and fusarium”. Please check the text carefully.

14. Page 15, L420-426. These two paragraphs can be merged into a single paragraph.

15. Many places are not italicized, please check the full text carefully.

16. Page 7, L250. It is suggested to change “3. Results of the study” to “3. Results and discussion

17. The bar charts in the article are too simple, all bar charts are re-beautified.

18. Please beautify Figure 10 and Figure 11.

Author Response

Thank you very much for your valuable review.

Our response is attached. 

Round 2

Reviewer 2 Report

Comments and Suggestions for Authors

The authors responded to most of the proposed changes. I'm not accept the answer to the comment 9.

9. Formula 3. Comments the same: Authors counted ungerminated seeds and P not determined seed germination. Response Thank you for your comment. There was a typo and the correction has been made: “Seed germination was determined on the tenth day, when analysing using formula 3: ? = ? × 100% ? (3) where, P - seed germination (%), n - number of ungerminated seeds on the tenth day, N - total number of plants in the sample.” 

If the value of ungerminated seeds is used in the formula, the result cannot refer to the germination index.

I also suggest changing the marks in the formula. Using the identical marks for  P - seed germination (formula 3) and P - percentage of infected plants (formula 4) will be misleading for potential readers.

Author Response

Thank you very much for the very useful review.

Our response is attached. 

Reviewer 3 Report

Comments and Suggestions for Authors

Nice work for the revision.

Author Response

THank you very much for your time and consideration.